# Assessing the impact of hydrodynamics on large-scale flood wave propagation – a case study for the Amazon Basin

Jannis M. Hoch[1,2], Arjen V. Haag[2], Arthur van Dam[2], Hessel C. Winsemius[2], Ludovicus P.H. van Beek[1], Marc F.P. Bierkens[1,2]

[1] Department of Physical Geography, Utrecht University, P.O. Box 80115, 3508 TC, Utrecht, The Netherlands
[2] Deltares, P.O. Box 177, 2600 MH, Delft, The Netherlands

*Correspondence to*: Jannis M. Hoch (j.m.hoch@uu.nl)

**Abstract.** Large-scale flood events often show spatial correlation in neighbouring basins, and thus can affect adjacent basins simultaneously, as well as result in superposition of different flood peaks. Such flood events therefore need to be addressed with large-scale modelling approaches to capture these processes. Many approaches currently in place are based on either a hydrologic or a hydrodynamic model. However, the resulting lack of interaction between hydrology and hydrodynamics processes, by for instance implementing groundwater infiltration on inundated floodplains, can hamper modelled inundation and discharge results where such interactions are important. In this study, the global hydrologic model PCR-GLOBWB at 30' spatial resolution was one-directionally and spatially coupled with the hydrodynamic model Delft3D Flexible Mesh (FM) for the Amazon River Basin at a grid-by-grid basis and at daily time step. The use of a flexible unstructured mesh allows for fine-scale representation of channels and floodplains, while preserving a coarser spatial resolution for less flood-prone areas, thus not unnecessarily increasing computational costs. In addition, we assessed the difference between a 1D-channel/2D-floodplain and a 2D schematization in Delft3D FM. Validating modelled discharge results shows that coupling PCR-GLOBWB to a hydrodynamic routing scheme generally increases model performance compared to using a hydrodynamic and hydrologic model only for all validation parameters applied. Closer examination shows that the 1D/2D schematization outperforms 2D for $r^2$ and RMSE whilst having a lower KGE. We also found that spatial coupling has the significant advantage of a better representation of inundation at smaller streams throughout the model domain. A validation of simulated inundation extent revealed that only those set-ups incorporating 1D channels are capable of representing inundations for reaches below the spatial resolution of the 2D mesh. Implementing 1D channels is therefore particularly of advantage for large-scale inundation models as they are often built upon remotely sensed surface elevation data which often enclose a strong vertical bias, hampering downstream connectivity. Since only a one-directional coupling approach was tested, and therefore important feedback processes are not incorporated, simulated discharge and inundation extent for both coupled set-ups is generally overpredicted. Hence, it will be the subsequent step to extend it to a two-directional coupling scheme to obtain a closed feedback loop between hydrologic and hydrodynamic processes. The current findings demonstrating the potential of one-directionally and spatially coupled models to obtain improved discharge estimates form

an important step towards a large-scale inundation model with a full dynamic coupling between hydrology and hydrodynamics.

## 1 Introduction

Global flood risk is increasing at an accelerating rate due to a combination of changed climatic conditions and intensified urbanization in proximity to rivers (Ceola et al., 2014; Hirabayashi et al., 2013; Jongman et al., 2012; Winsemius et al., 2015). This is reflected by a significant increase in economic losses in the latter half of the 20[th] century associated with flooding. In 2012 alone, economic losses exceeded $19 billion, comprising one third of all losses due to natural hazards (Munich Re, 2010; UNISDR, 2011; Visser et al., 2012). To better understand current and future hazard and risk, and to facilitate robust climate change adaption and mitigation measures, this study aims to show the strengths, weaknesses, and opportunities of spatially coupled hydrologic-hydrodynamic models compared to mere hydrologic and hydrodynamic models, respectively. We believe that coupling models is a pivotal corner stone for more realistic, robust, and integrated flood hazard and risk assessments.

Recently, modelling flood hazard and risk experienced a boost in attention as flood hazard maps are paramount for sound flood risk assessments (Hagen and Lu, 2011). In many cases, however, flood hazard maps are computed for geographically limited areas only. Because flood waves show strong spatial correlation in different but neighbouring basins, they can be considered to be large-scale phenomena, and, in turn, demand large-scale modelling approaches (Jongman et al., 2014), especially over data-scarce areas (Ward et al., 2015). The outcome of such large-scale models may be beneficial for global stakeholder as the United Nations Office for Disaster Risk Reduction (UNISDR) or the World Bank to, for instance, facilitate discussions with stakeholder's risks, better allocate their funding, but also for re-insurance companies or governmental entities (Ward et al., 2015). Tiling small-scale maps from different small-scale studies to obtain the required large-scale estimates is not a viable alternative as it introduces many sources of uncertainty and inconsistencies (Pappenberger et al., 2006, 2012) and does not account for any spatio-temporal correlation. Recent studies aimed to model large-scale flood hazard by dividing the model domain into various catchments (Alfieri et al., 2014; Dottori et al., 2016; Sampson et al., 2015). Notwithstanding the promising results, such approaches still require upstream boundary forcing, additional efforts due to division and merging, and still cannot fully account for the aforementioned spatial correlation of flood events in neighbouring basins as they use synthetic flood events.

Triggered by an increase in computational capacities and in availability of remotely sensed data for parameterization, calibration, and validation, research on large-scale inundation modelling was intensified in past years. For example, a range of global data sets is by now freely available such as, *inter alia*, Digital Elevation Maps (DEM) (e.g. HydroSHEDS, Lehner et al. (2008); ASTER; GTOPO30), water body maps (e.g. G3WBM, Yamazaki et al. (2015)), global river width and depth (Andreadis et al., 2013) or observed river discharge (Global River Discharge Centre (GRDC); Global River Discharge Project, (RivDIS)). In addition, algorithms to compute river widths globally (Yamazaki et al., 2014), to quantitatively

describe topography ("Height Above Nearest Drainage" (HAND), Rennó et al. (2008)) or to apply surface reconditioning (Yamazaki et al., 2012a) were presented.

With these data sets and algorithms being available, large-scale flood hazard modelling approaches are strongly facilitated. Most of the approaches can be categorized by (a) the processes represented and (b) the model schematization. While the latter category comprises possible schematizations such as 2D grid, 1D channels or coupled 1D/2D models, the first contains the possibility to in- or exclude several hydrologic or hydrodynamic models or their components in the computational backbone.

Global hydrologic models (GHM) such as PCR-GLOBWB (van Beek and Bierkens, 2008), WaterGAP (Döll et al., 2003) or VIC (Liang et al., 1994; Wood et al., 1992) are capable of modelling water balances, and, hence, available surface water volumes at the global scale. Another advantage is that hydrologic models can easily be forced with ensembles of Global Climate Models (GCM) which is beneficial for predictions of future changes in flood hazard and risk (Hirabayashi et al., 2013; Jongman et al., 2014; Weiland et al., 2010; Winsemius et al., 2015). However, large-scale hydrologic models strongly depend on the quality of their input data and robustness of their process descriptions, which may differ remarkably between individual catchments (Kling et al., 2015; Li et al., 2015). Besides, many GHM are relatively coarse scale, with the finest spatial resolution for global models currently being 5 arc minutes or 10 km x 10 km at the Equator (Bierkens, 2015). This may, although sub-grid post-processing can be used to meliorate outcomes as done for instance in the "Global Flood Risk with IMAGE Scenarios (GLOFRIS)" framework by Winsemius et al. (2013), reduce model accuracy since important floodplain properties and channel-floodplains dynamics can only be implemented in a simplistic manner.

Dedicated hydrodynamic models, on the other side, put their emphasis on the correct simulation of surface water flow and levels, and, hence, consider important factors such as inertia terms of channel geometry in more detail than most large-scale hydrologic models, as the latter often employ kinematic wave or Muskingum-Cunge approaches only. Thus, hydrodynamic models allow for simulating back-water effects which are pivotal flood triggering processes (Moussa and Bocquillon, 1996; Paiva et al., 2013). Hydrodynamic models are usually forced with upstream boundary conditions based on regionalization of observation stations (Huang et al., 2014; Sampson et al., 2015; Wilson et al., 2007). Yet, using observed boundary conditions makes them highly dependent on the presence and spacing of the stations. The aforementioned spatial correlation of flood waves can thus not realistically be modelled as important spatially-distributed flood triggering processes such as precipitation events over large surface areas would not necessarily be captured by the stations, as for instance the ENSO phenomenon in the Amazon River Basin (Molinier et al., 2009).

Most hydrodynamic modelling approaches are implemented by employing 1D, 2D or 1D/2D schematizations. Mere 1D models, however, have difficulties with modelling surface flow over larger areas and floodplains specifically, while regular 2D models inevitably lead to an increase in required computational power, especially if results need to be computed at a fine spatial resolution (Finaud-Guyot et al., 2011; Liu et al., 2015). In addition, 2D models experience problems in case the actual river width is smaller than the grid size and also in case there are multiple rivers within one cell, although it is possible to partly overcome that by applying sub-gridding routines (Neal et al., 2012). Besides, flow resistance to surface roughness is

overestimated in 2D set-ups. In addition to the currently employed techniques, use of flexible meshes is emerging which allows for both a fine spatial resolution in more relevant areas while at the same time not unnecessarily increasing computational costs where only limited dynamics and changes are expected. Such flexible gridding over the model domain may moreover be a viable avenue to meet the debated grand challenge of hyper-resolution modelling (Bierkens et al., 2015;

Wood et al., 2011). Yet, the application of flexible meshes focussed so far mostly on oceanic and coastal computations (Chen et al., 2003; Muis et al., 2016) and less on the representation of rivers and floodplains, although studies corroborate its high potential (Castro Gama et al., 2013).

Based on this, a call for a more holistic large-scale modelling approach can be formulated. Coupling existing models may provide an advantageous way forward as the strengths of individual models are maintained and weaknesses compensated. In

fact, many studies already integrate various disciplines by model coupling, for instance hydrologic with atmospheric models (e.g. Senatore et al. (2015); Wagner et al. (2016)), with climate models (e.g. Butts et al. (2014); Zabel & Mauser (2013)) or with glacier models (e.g. Naz et al. (2014); Zhao et al. (2013)). To obtain information about inundation patterns, approaches to couple hydrology with hydrodynamics were already explored in previous studies, but either at the sub-catchment scale only (Paiva et al., 2013; Rudorff et al., 2014a, 2014b); by using a land surface model (LSM) to obtain input (Pappenberger et

al., 2012); by employing the hydrologic model VIC (Liang et al., 1994; Wood et al., 1992) to compute boundary discharge for LISFLOOD-FP (Bates and de Roo, 2000) in the Lower Zambezi River (Schumann et al., 2013); by using output from a hydrologic model as lateral inflow for LISFLOOD-FP to model inundation dynamics in the Ob River (Biancamaria et al., 2009); or by using used output from GloFAS ("Global Flood Awareness System") (Alfieri et al., 2013) with hydrodynamics to obtain synthesized floods with different return periods (Dottori et al., 2016). Notwithstanding the contributions of these

studies to current flood risk understanding, they still lack the capability to produce hydrological forcing within the actual model domain, and are thus not able to simulate the feedback between hydrology and inundation processes on floodplains.

In the present study, we present a one-directional and spatially explicit coupling approach between the global hydrologic model PCR-GLOBWB and the state-of-the-art hydrodynamic model Delft3D Flexible Mesh, allowing for the exchange of information throughout the entire model domain. To our knowledge, this is a novelty in large-scale inundation modelling.

Moreover, the exchange of variables between hydrology and hydrodynamics takes place on a grid-to-grid basis at the time step or even sub-time step level. This approach allows for online coupling, thus providing the potential to eventually perform two-directional exchange of information. The Amazon River Basin was schematized with both a 2D flexible mesh and a 1D/2D set-up, allowing us to test potential (dis-)advantages between both set-ups. Additionally, the hydrologic and hydrodynamic models were also run in a stand-alone mode to fully assess the added value of model coupling. The utilization

of only global data sets and algorithms ensures transferability to other basins as well as a straightforward scalability of our approach to larger scales. It is moreover a part of the study's aim to detect the most suitable model set-ups to continue with future extensions and larger-scale applications of our coupling technique.

With our approach we are confident to close the gap between hydrology and hydrodynamics, and to make a step towards a global fully-fledged inundation model. Such a model set-up can provide information on spatial correlations and interrelations

between flood events, ultimately facilitating current large-scale flood hazard and risk assessments. Eventually, this can be used for the formulation of more robust climate change adaption and mitigation measures, and to further inform global flood risk policies.

## 2 Methodology

The two models used for this study are the global hydrologic model PCR-GLOBWB (van Beek, 2008; van Beek et al., 2011), and the state-of-the-art hydrodynamic model Delft 3D Flexible Mesh (FM) (Deltares, 2016; Kernkamp et al., 2011). To test the added value of our coupling approach as well as the differences between 2D and 1D/2D schematization, the following experimental set-up was designed, consisting of five modelling runs: (i) PCR-GLOBWB with its DynRout-extension to obtain purely hydrology-based results; (ii) and (iii) a 2D and 1D/2D Delft 3D FM schematization both forced

with discharge observed at GRDC stations to obtain purely hydrodynamic-based results; (iv) and (v) the same two FM-schematizations forced with output from PCR-GLOBWB. For all runs with Delft 3D FM, a constant water level of 0.0 m is assumed at the river mouth as a downstream boundary. Even though the influence of ocean tides is reported to be significant (Lima et al., 2003), tidal dynamics were not considered in the present study as it exceeds the scope of the work.

Each set-up was applied for the Amazon Basin for the period from 1$^{st}$ of January 1985 until 31$^{st}$ December 1990. This early

period had to be chosen as for some GRDC-stations no more recent discharge data is available. Output of all cases was validated against observed GRDC discharge data at Óbidos (GRDC Station Nr. 3629000), the most downstream GRDC-station available (Figure 1). To this end, three functions were applied for validation: the coefficient of determination ($r^2$) to assess the reproduction of the shape of the hydrograph; the Root Mean Squared Error (RMSE) to assess the water balance; and the Kling-Gupta Efficiency (KGE) (Gupta et al., 2009) to evaluate the model's skill. In addition, we qualitatively

inspected the inundation extent and water levels for the various model runs. We employed LandSat imagery taken on July 1$^{st}$ 1989 to validate simulated inundation extent, as it is one of the few cloud free images at this time, and represents inundation patterns during peak season. To compare simulated water levels, four observation points on floodplains along the main river reach were defined (Figure 1): "Loc1" close to the delta (1.62° S, 52.46° W); "Loc2" downstream of Óbidos (2.15° S, 54.55° W); "Loc3" just upstream of Óbidos (2.45° S, 56.81° W); "Loc4" even further upstream (2.97° S, 58.35° W).

**2.1 The hydrologic model: PCR-GLOBWB**

To generate hydrologic input, the global hydrologic model PCR-GLOBWB at 30' resolution (approximately 55 km x 55 km at the Equator) was applied. It is entirely coded in PCRaster Python (Karssenberg et al., 2010). PCR-GLOBWB distinguishes between two vertically stacked soil layers, an underlying groundwater layer, and a surface canopy layer. Water can be exchanged vertically, and excess surface water can be routed horizontally along a Local Drainage Direction (LDD)

network. In the present study, the kinematic wave approach was used for routing, and Manning's surface roughness coefficient was uniformly set to 0.03 s m$^{-1/3}$. This value is in line with other studies in the Amazon Basin (Paiva et al., 2013;

Rudorff et al., 2014a, 2014b; Trigg et al., 2009; Yamazaki et al., 2011). A uniform value was chosen to eliminate this factor as a cause for differences when comparing the stand-alone runs as well as 1D/2D set-ups. The model was forced with CRU precipitation and temperature data (Harris et al., 2014), and evaporation was computed using the Penman-Monteith equation. Data sets were downscaled to daily fields for the period from 1957-2010 using ERA40/ERAI (Kållberg et al., 2005; Uppala et al., 2005). For more information on PCR-GLOBWB we refer to van Beek & Bierkens (2008) and van Beek et al. (2011). PCR-GLOBWB was already applied in various studies: Weiland et al. (2010) investigated how forcing from different global circulation models can reproduce global discharge variability; Yossef et al. (2012) concluded that PCR-GLOBWB shows skill when used for flood forecasting; and Wanders & Wada (2015) employed the model to assess the impact of humans and climate on drought in the 21$^{st}$ century; de Graaf et al. (2015) fully coupled PCR-GLOBWB with a physically-based groundwater model capable of simulating lateral flows.

From a priori runs we were informed that PCR-GLOBWB underestimates discharge in the Amazon Basin. To eventually obtain discharge values that are close to observed values and enhance the significance of the validation procedure, we therefore decided to apply a simplistic regional optimization technique for five model parameters. To this end, we tested the model's performance sensitivity to a range of multipliers for these parameters, using the log-scaled Nash-Sutcliffe coefficient of simulated discharge at Óbidos as performance indicator. Based on performance, we then chose the combination of multipliers resulting in the highest log-scaled Nash-Sutcliffe coefficient. Consequently, the minimum soil depth fraction for which interflow is calculated, the log-scaled saturated hydraulic conductivity of groundwater flow ($k_{sat}$), and the log-scaled recession coefficient were multiplied by 0.5. The general parameterization of PCR-GLOBWB, however, remained unaffected and no further local calibration was performed to preserve the global applicability of the model. PCR-GLOBWB also has the option to include human water use from irrigation, households, and industry as an integral part of its model runs. In our application however, we decided to simulate river discharge under natural flow conditions.

**2.2 The hydrodynamic model: Delft3d Flexible Mesh**

For hydrodynamic calculations, the state-of-the-art model Delft 3D Flexible Mesh (FM) was employed (Kernkamp et al., 2011). It allows the user to schematize the model domain with a flexible mesh in 1D/2D/3D, and therefore supports the computationally efficient schematization of topographically challenging areas such as river bends or irregular slopes. The model solves the full Saint-Vernant equations, or shallow-water equations (SWE). Solving the SWE is, as stated before, a major advantage compared to most large-scale hydrodynamic and hydrologic models because this is essential to account for important flood-triggering processes such as back-water effects (Moussa and Bocquillon, 1996; Paiva et al., 2013). In analogy to PCR-GLOBWB, the surface roughness coefficient was set to 0.03 s m$^{-1/3}$ to guarantee comparability. We expressly desisted from calibrating model parameters for any of our hydraulic model set-ups due to two reasons. First, calibration may obscure the actual performance of the model set-ups with respect to real governing hydraulic processes as their quality may possibly be governed by calibration. And second, we aim to apply the presented coupling scheme at other

basins. Locally calibrating FM for the Amazon Basin may introduce inconsistencies among the global datasets used for model setup, and jeopardize their validity for ungauged basins.

Due to its very recent publication, only a limited number of published studies using Delft 3D FM are available. It was, for instance, applied in a global-scale reanalysis for extreme sea levels (Muis et al., 2016). In another study, Castro Gama et al.

(2013) applied Delft 3D FM successfully to model flood hazard at the Yellow River, and concluded that applying a flexible mesh reduces computation time by a factor 10 compared to square grids with equal quality of model output.

## 2.3 Defining the 1D network

The course of the 1D river channels as well as effective river width $w$ were derived based on the "Global Width Database for Large Rivers (GWD-LR)" algorithm by Yamazaki et al. (2014), hence already accounting for river braiding and islands.

Comparing both the course and the computed width of the obtained 1D network schematization with OpenStreetMaps (OSM) yielded an overall good fit with lower goodness-of-fit in meandering and delta regions (Figure 2).

River depth $d$ [m] was subsequently estimated from river width $w$ [m] by combining the following equations from Paiva et al. (2011), with $A_d$ [km$^2$] being the upstream area of one point along the river:

$$w = 0.81 A_d^{0.53} \tag{1}$$

$$d = 1.44 A_d^{0.19} \tag{2}$$

to the following width-depth relation:

$$d = 1.55 w^{0.36} \tag{3}$$

Benchmarking the resulting river depths obtained with Eq. (3) with those found in a global river bankfull width and depth database by Andreadis et al. (2013) showed better results than those obtained with the widely used width-depth relation

proposed by Leopold & Maddock (1953). By means of the aforementioned equations a maximum depth of 54 m, a minimum depth of 5 m, and an average depth of 13 m were computed. Finally, width and depth information was stored in cross-sections along the network with a spacing of around 20 km (Figure 2). We are confident that applying a hydro-geomorphic relation between river depth and observed width, as also applied by Neal et al. (2012), is valid in this case due to three reasons. First, the equations were constructed based on a large sample of cross-section information, and thus can be

extrapolated over larger areas of application. Second, we consider the width information of GWD-LR to reflect local conditions better than the more one-dimensional approach of relating bathymetric information on only one predicting variable such as for instance upstream area or discharge. Last, bathymetric information is internally interpolated in the model, preventing any extreme local variations in observed river width and depth, consequently also avoiding improbable local flow hindrance.

## 2.4 Defining the 2D flexible mesh

For surface elevation values, we used the HydroSHEDS data set, which was derived from the Shuttle Radar Topography Mission (SRTM) (Lehner et al., 2008). Because significant vertical measurement errors emanate from the C-band Synthetic Aperture Radar (SAR) used by SRTM, extensive hydrologic conditioning was carried out in this study to remediate the most
relevant errors in currently available data sets.

First, noise by vegetation cover was reduced. This is essential as the radar signal cannot fully penetrate dense canopy, leading to quality degradation especially in rainforests (Berry et al., 2007). As a result, absolute vertical errors of around 22 m were found in the Amazon Basin (Carabajal and Harding, 2006; Sanders, 2007). The approach used in the present study to account for vegetation cover is described in detail by Baugh et al. (2013). For the present study, 50 % of the canopy heights
reported by Simard et al. (2011) were subtracted from original elevation values as proposed by Baugh et al. (2013).

Even after vegetation was removed, flow connectivity can be hindered by grid cells surrounded by higher elevated cells which can stem from elevation irregularities such as islands, bridges or other residues. Thus, these local depressions were removed in a second step to guarantee downstream flow connectivity along flow paths. Conventional procedures such as lifting downstream cells or stream burning fail however to adequately address this issue as the land surface is altered one-
sided, and should not be applied to rivers in flat environments such as the Amazon River (Getirana et al., 2009). Hence, a more advanced algorithm based on the work of Yamazaki et al. (2012) was applied. This algorithm either 'digs' or 'fills' along a flow path as defined by the HydroSHEDS LDD, resulting in smoothened elevation values along downstream flow paths as demonstrated for two flow paths in .

While for 1D/2D applications, the 1D vector channel data is embedded into the smoothed 2D elevation, it was necessary to
compute bathymetric information for the 2D schematizations. This is because the DEM used lacks reliable information about river bathymetry as the SRTM radar signal is not able to fully penetrate deeper water bodies. To derive bathymetry information, current research projects aim to exploit available remotely sensed data or aerial photography (Kinzel et al., 2013; Legleiter, 2015, 2016; Yoon et al., 2012). Yet, obtaining satisfactory information for large-scale river bathymetry remains a major research challenge. For the present study, river depth $d$ was computed as a function of upstream area $A_d$ as
follows: for all grid cells where $A_d \geq 10^4$ km$^2$, Eq. (2) was applied to compute $d$ on a grid-by-grid basis. The threshold of $10^4$ km$^2$ was chosen after trial-and-error to filter many small and short reaches which were not represented by the 1D network. Due to the differences in 1D vector network and Local Drainage Direction (LDD) map used for the 2D raster data, it was however not possible to precisely apply the same equations. Despite these minor differences in methodology, manual inspection of computed river depths computed for 1D channels and 2D bathymetry, revealed no major discrepancies in our
model domain, and we therefore consider both ways to compute bathymetry valid, particularly in light of the limited availability of bathymetry data for large-scale applications. The computed depth of one specific pixel was then spread to all cells whose distance is shortest to the pixel under consideration. Subsequently, the resulting bathymetry map was created by lowering elevation values of only those pixels defined as permanent water bodies in the Global 3-second Water Body Map

(G3-WBM) developed by Yamazaki et al. (2015) (see Figure 2). The computed elevation values were subsequently interpolated over the flexible mesh, and elevation values per FM-cell are obtained by unweighted spatial averaging of the computed elevations at the cell vertices.

Since the hydrodynamic computations and model coupling still require significant computational power for multi-year simulations, the modelling domain of Delft 3D FM was limited to flood-prone areas. To derive a suitable extent, the Height Above Nearest Drainage (HAND) algorithm was applied (Rennó et al., 2008) as it yields relative terrain elevation to the nearest hydrologically connected drainage. The flexible mesh was then obtained by automatic local grid refinement of a coarser regular grid based on the obtained HAND values and limiting it to grids where computed HAND values are less or equal to 25 m, that is until terrain reached an elevation of 25 m above the nearest water body. The final model domain is presented in Figure 1 and still encompasses an area of around $1.2 \times 10^6$ km$^2$ which is nearly a fifth of the entire Amazon River Basin. The threshold was chosen arbitrarily but model results showed that it is sufficiently large. By establishing the refinement on this algorithm, the flexible mesh has the finest spatial resolution (2.5 km $\times$ 2.5 km) for areas with lowest HAND values, such as water bodies and floodplains, while areas with higher HAND values, and hence areas more remote from water bodies, are modelled with coarser spatial resolution up to 10 km $\times$ 10 km per grid. In these latter regions, the number of grid cells is thus reduced by a factor of 16, benefitting the stability-limited computational time step and significantly reducing overall computation times.

## 2.5 Coupling the models

Coupling PCR-GLOBWB with Delft 3D FM was achieved by means of the Basic Model Interface (BMI). Peckham et al. (2013) proposed the BMI as a tool within the Community Surface Dynamics Modeling System (CSDMS) project to exchange information between separate models at any given time step. By exposing certain internal state variables of the model by means of the BMI, interactive modelling is facilitated as these variables can be modified during the model execution.

Generally, each BMI has several functions that can be called from external applications like, as in this case, a Python script. First, models need to be initialized. Second, the BMI enables the user to retrieve variables, and to manipulate them if required, for instance to convert units or to add values. Third, the manipulated variables can be set back to the original model or can be used to overwrite variables in one or multiple other models, given that they agree to the internal data structure of those models. Fourth, models connected to a BMI can be updated at a user-specified time step. This way it is possible to get, change, and set variables during the execution of the models in use. In a last step, models can be finalized to end the computations. It has to be noted that for each model involved one specific BMI adapter has to be developed with respect to the specific internal model structure and programming language. Whilst PCR-GLOBWB is already in Python and its BMI implementation is hence straightforward, Delft 3D FM offers a native C-compliant BMI-implementation which can be called from within Python using the BMI-python package (see https://github.com/openearth/bmi-python). For further information on the BMI, we refer to Peckham et al. (2013) and the related website (CSDMS, 2016).

In order to be able to spatially couple both models, it is required to overlay the model extent of both FM and PCR-GLOBWB. To this end, the centroid of each 2D FM-cell was computed, and a FM-cell is then considered to be coupled to PCR-GLOBWB if its centroid is located within the bounds of the PCR-cell. The coupling algorithm (Figure 4) was employed at a daily time step: first, PCR-GLOBWB was run for one day, then, a daily delta volume, that is the volume to be added to FM with the day's timestep, was computed for every coupled PCR-cell as the sum of daily river discharge inflows at the boundary of FM, and local surface runoff throughout the model domain. The daily delta volume was subsequently divided over and added to all FM-cells within this specific PCR-cell. Note that this explicit spatial forcing of Delft3D FM is fundamentally different from the GRDC-fed runs, where only upstream discharge boundary conditions are applied, and no spatially distributed forcing is active. As only the most downstream part of the Amazon Basin is schematized in FM, no coupling was performed for the upper part of the basin. For these un-coupled areas, PCR-GLOBWB is run stand-alone, and water is routed towards the coupled domain using the kinematic wave approximation. Within the coupled area, the LDD of PCR-GLOBWB was deactivated to prevent further routing in the hydrologic model. As a last step in the coupling algorithm, FM was updated and integrated forward in time until it reaches the same model time step as PCR-GLOBWB to compute daily inundation and discharge values. Since only a one-way coupling approach is tested, water added to FM can only be routed downstream, but cannot infiltrate or evaporate, most likely leading to overestimation of modelled discharge and inundation.

## 3 Results and discussion

### 3.1 Discharge simulation at Óbidos

PCR-GLOBWB-DynRout reproduces low flows well, but fails in reproducing the observed variation in discharge as shown by a low coefficient of determination (Table 1). This low value can be attributed to the rugged hydrograph obtained, as shown in Figure 5. The strong fluctuations cannot be fully explained, but we assume that they may be related to the simplistic routing scheme used, as discharge results for the coupled run do not show such behaviour, although they receive the same hydrologic input. In addition, peak discharge is generally modelled too early. This low performance is related to PCR-GLOBWB-DynRout being a global hydrologic model, thus not specifically designed for simulating discharge at the basin-scale despite the regional optimization technique applied for this study. The employed kinematic wave approximation as well as the coarse resolution of 30' can be identified as factors currently hampering a more accurate simulation of discharge.

Forcing the model with discharge observed at GRDC-stations, we found that the aggregated input discharge as obtained from upstream GRDC-station observations (Figure 1) accounted for only 59% of the discharge generated in the basin as observed at Óbidos (Figure 6). This underrepresentation can be linked to the discrepancy between catchment area at Óbidos and summed catchment area of all input stations upstream of Óbidos. Comparing both, we found that only 63% of upstream catchment area at Óbidos is accounted for by input stations (Table 2). The differences in discharge can therefore be

attributed to the additional discharge created in the intermediate area between Óbidos and the upstream inflow stations. To avoid the expectable too low discharge estimates and facilitate comparability with other model runs, we therefore decided to scale the input discharge values accordingly. The results then reveal that the strength of purely hydrodynamic runs is the correct reproduction of discharge variability, as shown by high coefficients of determination. Still, model results obtained

with only Delft 3D FM resulted in lagged discharge, with the 1D/2D schematization having lower discharge results and a larger time lag. We suspect that the obtained attenuation and time lag for both 2D and 1D/2D schematization result from the absence of any internal forcing. By using only upstream discharge boundaries and neglecting internal sources, discharge will need longer to propagate until Óbidos due to the larger average travel distance. It should be noted that from a computational point of view, the 1D/2D set-up has the advantage of a 25% lower wall clock time required to finish the simulation period

compared to the 2D set-up.

Assessing model results for the coupled runs, we see that the simulated discharge is higher than of both the purely hydrology- and purely hydrodynamic-based models. Deviations between coupled and GRDC-runs can be ascribed to differences in forcing, which is not only different in terms of input volumes, but also in terms of input locations. We also find that the coupled runs do not reach the same variability in discharge as the GRDC-forced runs, although employing the

same model schematizations. This may be related to a higher proportion of overland flow resulting from distributing water volumes over the FM-cells, which would reduce discharge dynamics. The disparities in discharge of coupled runs compared to PCR-GLOBWB-DynRout, however, have to be attributed to a combination of differences between model schematizations and process representation as we have carefully examined the water balance throughout the entire coupling process, and therefore can exclude volume errors as source of deviations. First, Delft 3D FM and PCR-GLOBWB-DynRout differ in their

spatial resolution, with the latter having a much coarser spatial resolution. Eventually, this difference can have an impact on modelled discharge accuracy, because the role of channel-floodplain interaction is pivotal for inundation and discharge estimates and so is schematization of connecting channels (Neal et al., 2012; Rudorff et al., 2014a; Savage et al., 2016) which both are facilitated by using finer spatial resolution. This is underlined by the smoothened daily discharge which result when replacing the simple kinematic wave routing at 30' spatial resolution with a hydrodynamic model at fine spatial

resolution, even though both are subject to the same meteorological forcing as well as hydrologic processes. Second, differences in process description can lead to improved discharge estimates compared to PCR-GLOBWB-DynRout. In particular, solving the SWE – as implemented in Delft 3D FM – instead of the kinematic wave approximation may have influenced results as it accounts for back-water effects which play an important role in the Amazon Basin because of its low gradients (Meade, 1991; Moussa and Bocquillon, 1996; Paiva et al., 2013). And third, our coupled set-ups may yield higher

discharge than PCR-GLOBWB-DynRout due to the one-directional coupling scheme implemented. For peak flow conditions, the higher discharge can be attributed to the absence of important groundwater infiltration and evaporation processes on inundated areas, resulting in increased surface water volumes routed downstream. Note that in PCR-GLOBWB-DynRout flooded areas are subject to evaporation which can partly explain the higher discharge resulting from the one-directionally coupled model. During low flow conditions however, the excess water that remained on the

floodplains, although it should have infiltrated or evaporated, can return into the channel, resulting in higher discharge too. Comparing our results to other studies, we find that both coupled runs have remarkably lower RMSE than those reported in Alfieri et al. (2013) for GloFAS. The obtained coefficients of determination come close to those by Yamazaki et al. (2011) and Yamazaki, Lee, et al. (2012), who connected runoff from a land surface model with a river-floodplain routing scheme.

**3.2 Water level and extent of inundation**

Assessing modelled inundation water levels, we find that, because discharge results are almost identical, simulated water levels for the GRDC-fed runs differ only slightly between 2D and 1D/2D schematization, with the latter generally showing lower water levels (Figure 7). This is the result of the 1D channels providing better hydraulic connectivity throughout the study area since also smaller side channels below the spatial resolution of the 2D mesh are accounted for (Figure 8). Results
furthermore show that for some observation locations, the GRDC-runs yield higher water level values than 1way-coupled runs and vice versa at other locations. As the model schematizations are exactly the same, these local differences can be related to the difference in volume input into the FM model domain (dividing over FM-cells with PCR-GLOBWB output versus upstream boundaries with GRDC data), as well as local influence of precipitation events within the intermediate catchment area on water level dynamics. The discrepancy between simulated water level for 1D/2D and 2D set-ups at Loc2
exemplifies the impact vertical errors in input elevation data can have on 2D schematizations. While the area where the location was placed could be conveyed by the 1D network, this was not possible in the 2D set-up, thus resulting in local accumulation of water in a local depression. Results also indicate that locations closer to the delta (see Loc4 as an example) are less influenced by river dynamics or precipitation events, but more by the downstream water level boundary, for which smaller differences in simulated water level between model runs are revealed. From a holistic point of view, large-scale
water level dynamics are correctly represented with only minor differences between model set-ups, despite the results at Loc2 as mentioned above.

In terms of inundation extent, we performed a first-order and only qualifying validation of simulated against observed water extent for all runs except the DynRout set-up. Our results indicate that the 1D-2D schematization with GRDC-forcing performs particularly well (see Figure 8). This demonstrates the advantage of implementing 1D channels as inundation
extent is modelled more accurately, especially for smaller side-branches of the stream where the 2D resolution does not allow for detailed simulation of channel-floodplain interaction. This finding is in line with the observations made by Neal et al. (2012) who employed a sub-gridding scheme. For the coupled set-ups, water extent is well modelled for the main reaches of the river, but overpredicted for floodplain areas. We attribute these deviations to both the quality of remotely sensed input elevation and the coarse spatial resolution of the flexible mesh which may overly facilitate flow over floodplains. Besides,
distributing water volumes over the FM-cells in the coupling process may also have led to stronger inundation on floodplain areas than point inflow from GRDC-stations. Assessing simulated water extent over the entire study area, we again find that the use of 1D channels can highly improve the level of detail for river streams and bends for both the main branch as well as more remote areas, as shown in Figure 9. Similar as for the local water level validation, we found that the areas where

inundation is modelled differ strongly compared to the GRDC-runs. While inundation for those runs is limited to streams that are connected to upstream discharge boundaries, spatially coupling hydrology with hydrodynamics additionally yields inundation information for smaller reaches throughout the entire model domain which otherwise would not be fed with water. This constitutes a major improvement, and is a strong hint that model coupling can indeed contribute to better inundation extent estimates. Notwithstanding this achievement, we again see that water can accumulate locally which can partially be related to the presence of temporarily filled depressions during rainfall, and partially to the spatial resolution of the hydrodynamic model in combination with the quality of the elevation data used for model schematization. Also on the large picture, the local accumulation of water is less severe in the 1D/2D than in the 2D set-up due to a facilitated hydrologic connectivity within the river basin.

**4 Conclusion and recommendations**

In the present study, we spatially coupled the global hydrologic model PCR-GLOBWB with the state-of-the-art hydrodynamic model Delft 3D Flexible Mesh (FM), and compared resulting discharge and inundation extent with estimates obtained from stand-alone runs as well as actual observations to investigate possible strengths, weaknesses, and opportunities of model coupling for large-scale inundation modelling.

Our results showed that hydrology-only runs conducted with PCR-GLOBWB-DynRout have the least accurate discharge simulation of all runs. Particularly discharge variability could not be captured by a global hydrology model due to its coarse spatial resolution and its kinematic wave approximation of surface water flow in an area with limited topographic gradients. The question therefore remains: which is the most important, the coarse resolution or the simple hydrodynamics. Therefore, once PCR-GLOBWB at 5' spatial resolution is fully tested and available, the model runs should be repeated to better understand whether results can be improved by finer spatial resolution or are constrained by the employment of a kinematic wave approach. Besides, fine-tuning of sensitive parameters of PCR-GLOBWB at a global scale seems to be required to obtain a better-timed peak flow, not only for those optimized so far, but also others such as Manning's surface roughness coefficient.

Comparison revealed that runs forced with observed discharge from GRDC, once the underrepresentation of water volume in the systems was accounted for, outperform hydrology-based models in resembling discharge dynamics. While validation of GRDC-forced runs against observed discharge showed good performance, the disadvantage of such set-ups is the limitation of discharge to river reaches fed by the discharge boundaries. As a result, inundations along reaches that start within in the domain or along reaches not being fed by upstream discharge boundaries cannot be simulated. A first qualitative validation of simulated inundation extent with Landsat imagery showed that for those rivers connected to upstream discharge boundaries, the 1D-2D schematization with GRDC-forcing showed the best performance of all runs. Representation of 1D channels results in a better conveyance of surface water in the model domain and consequently less flood artefacts, in particular where 1D channel dimension is below the grid size of the 2D grid cells. We also found that GRDC-forced runs

show stronger attenuation and lagged peak discharge due to the longer average travel time required to propagate from the boundaries through the model domain.

Both 1D/2D and 2D coupled runs were able to capture the peak flow better than GRDC-runs, and to follow the discharge dynamics better than the simple kinematic wave model. The fact that they overpredict peak discharge for some years can be attributed to the absence of a feedback loop to hydrological processes on floodplains, such as groundwater infiltration and evaporation. It will be the aim of a follow-up study to implement a fully-dynamic coupling scheme, whereby information is exchanged between hydrology and hydrodynamics at each time step, and water on the floodplains is allowed to evaporate or recharge the groundwater store. We expect that this will lead to lower and more accurate discharge estimates. Replacing the simplistic routing scheme of PCR-GLOBWB with a full hydrodynamic model remarkably improves the coefficient of determination as well as the model's skill. From our results we conclude that spatially coupling hydrology and hydrodynamics merges the best of two worlds, namely water volume accuracy and routing scheme. From a computational point of view, the use of a 1D/2D set-up is favourable as it requires less computational time. At the same time it yields a better spatial resolution of the river network than the 2D set-up because it decreases dependency on quality of space-born DEM data sets which are known for introducing errors in large-scale inundation models. Especially for the coupled runs, these vertical errors are partly responsible in overestimated inundation extent and local water levels, in particular in floodplain regions. Another part of the overestimation may lie in the way water volumes are distributed over the 2D grid. It needs to be researched in more detail how the distribution of volumes impacts model results, and whether other techniques such as adding water directly into the 1D channels than onto the 2D grid may improve model performance. Besides, a future study should contain an assessment of the impact of varying spatial resolution of both the hydrologic and the hydrodynamic model as well as their interplay to obtain a better picture of the potential of model coupling at larger scales.

In this study, we used only global data sets for both the hydrological and the newly developed hydrodynamic model Delft 3D FM. Thus, the presented set-up can easily be applied in other river basins as well. On the long-term, we are confident that the proposed spatially coupled model set-up can eventually contribute to a better assessment of both current and also future flood hazard and risk.

**Author contribution**

A.V. Haag prepared the code for model coupling. A. van Dam supported the application of Delft3D Flexible Mesh and L.P.H. van Beek provided information for PCR-GLOBWB and PCR-GLOBWB-DynRout runs. L.P.H. van Beek, H.C. Winsemius and M.F.P. Bierkens supervised the research and provided important advice. J.M. Hoch designed and executed the research, as well as prepared the manuscript with thankful contribution from all co-authors.

## Acknowledgments

The authors declare that they have no conflict of interest. The authors want to acknowledge the valuable contributions of Gennadii Donchyts, Herman Kernkamp and Robert Leander from Deltares as well as Edwin Sutanudjaja from Utrecht University. This study was financed by the EIT Climate-KIC programme under project title "Global high-resolution database of current and future river flood hazard to support planning, adaption and re-insurance". We want to acknowledge the constructive contributions of one anonymous reviewer and Dr. Dai Yamazaki who helped to strongly increase the quality of the manuscript.

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

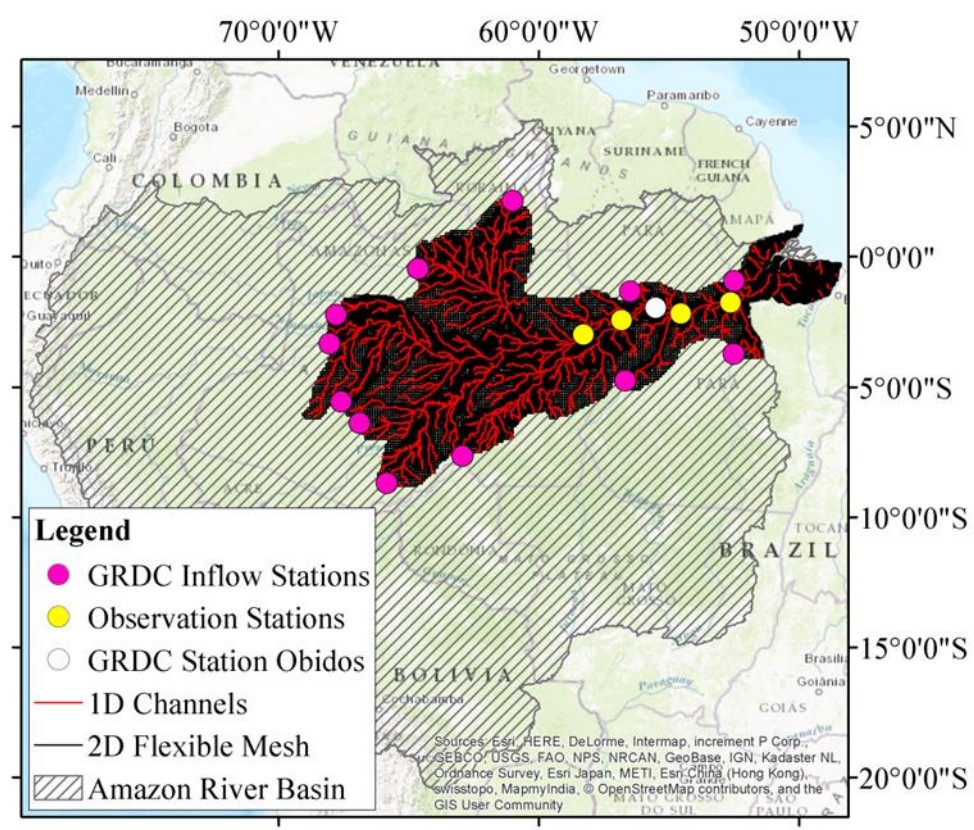

**Figure 1: Map of extent of 2D grid and 1D channels as part of entire Amazon River Basin; additionally shown the water level observation stations 1-4 counting from delta to upstream, as well as GRDC station Óbidos for discharge measurements and all GRDC input stations**

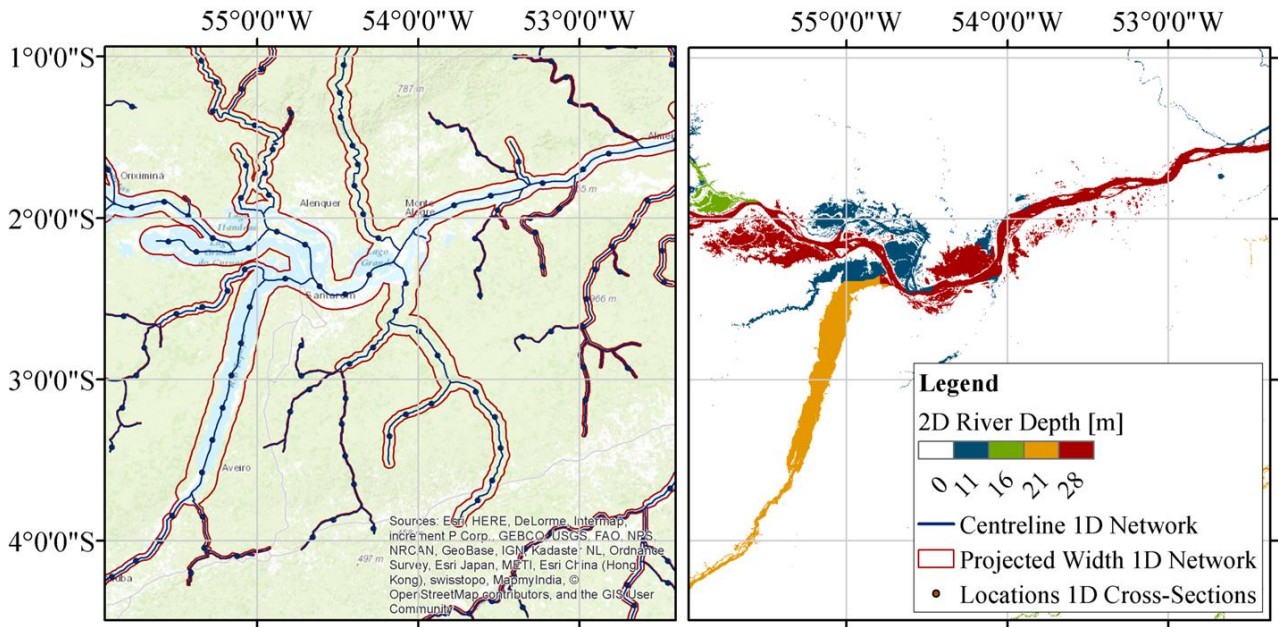

**Figure 2: 1D-network centre line, cross-section, and its computed (projected) width as well as computed river depth for all cells defined as permanent water bodies in G3WBM for same detail**

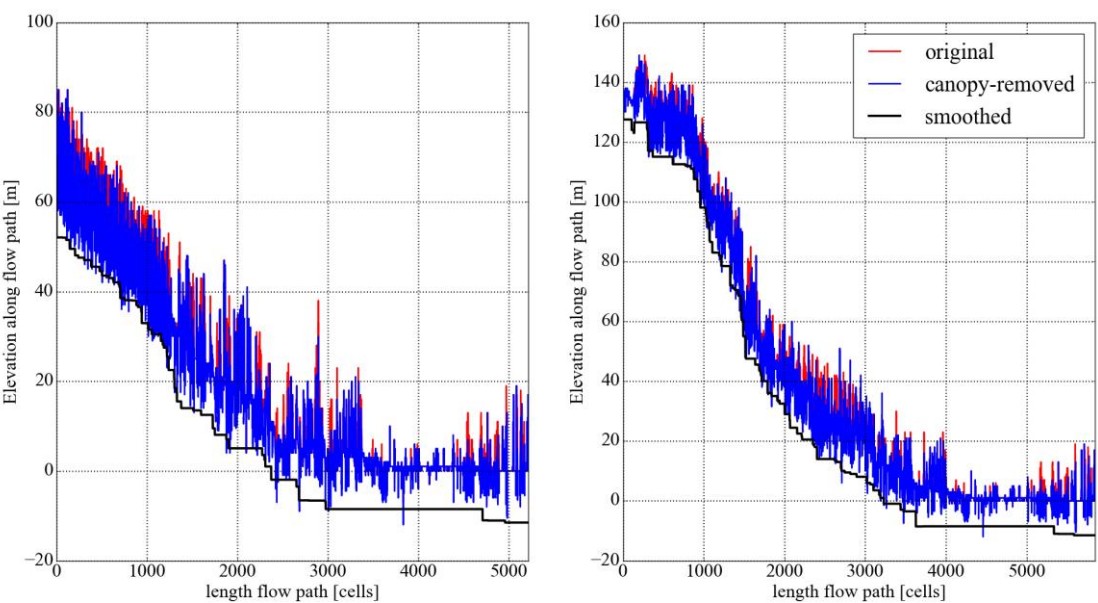

**Figure 3: Impact of vegetation removal ("canopy-removed") and surface reconditioning ("smoothed") on surface elevation along two exemplary flow paths compared to original HydroSHEDS-DEM data ("original")**

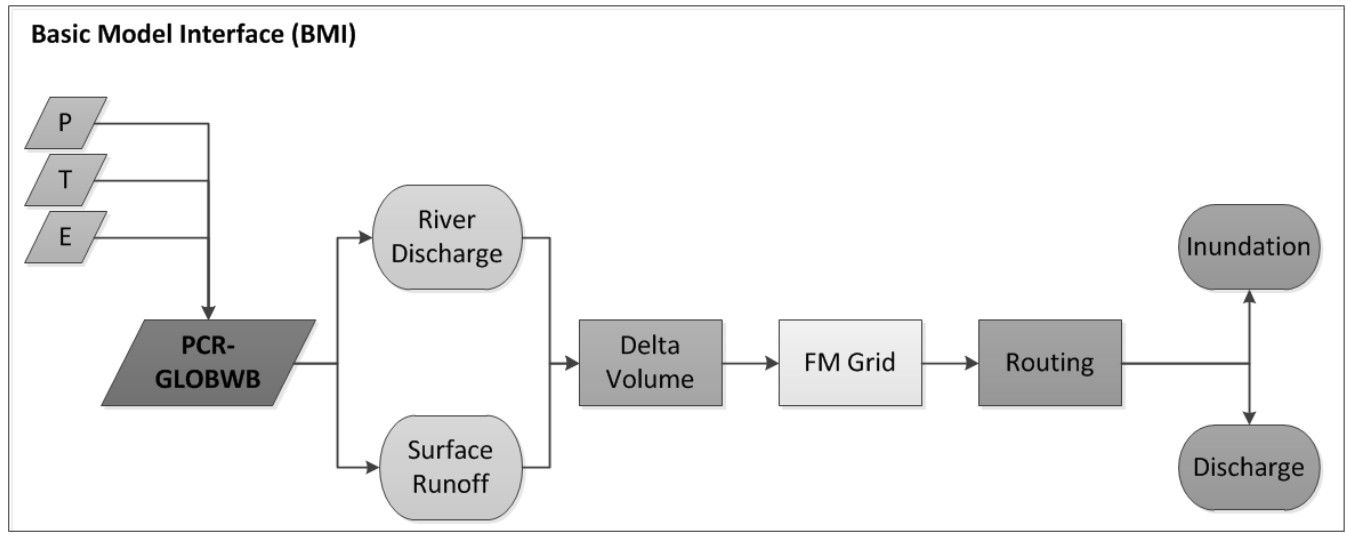

**Figure 4: Flow diagram of coupling process steps embedded in the Basic Model Interface**

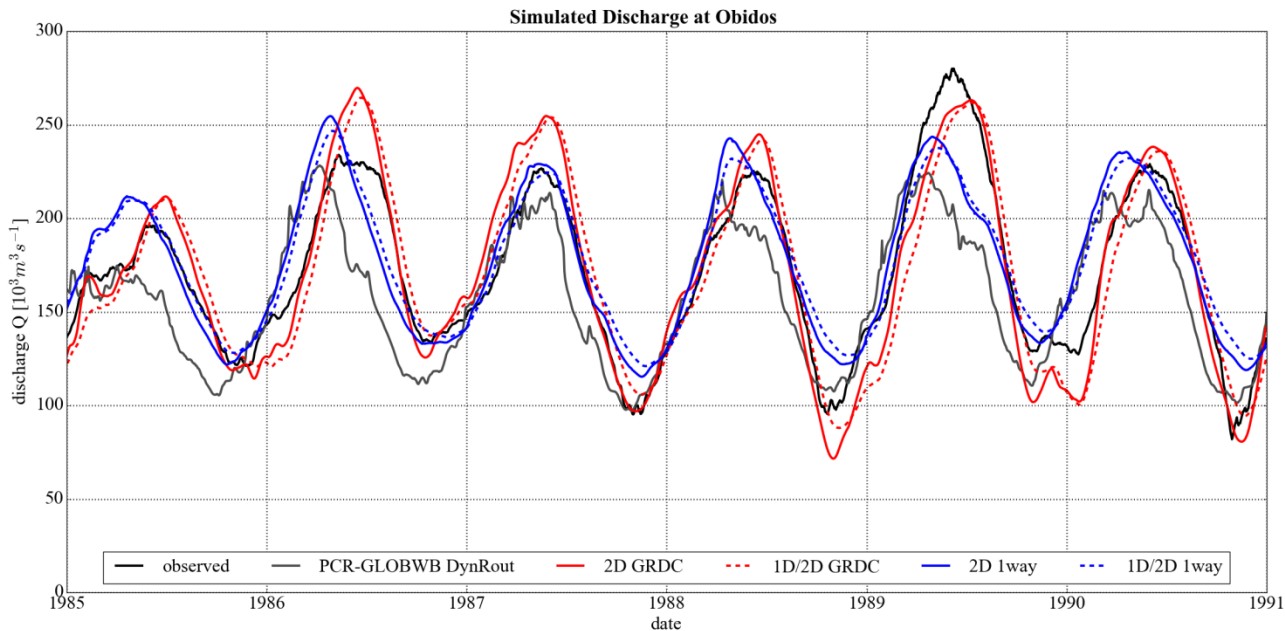

**Figure 5: Plot of all model results and observed discharge values at GRDC-station Óbidos**

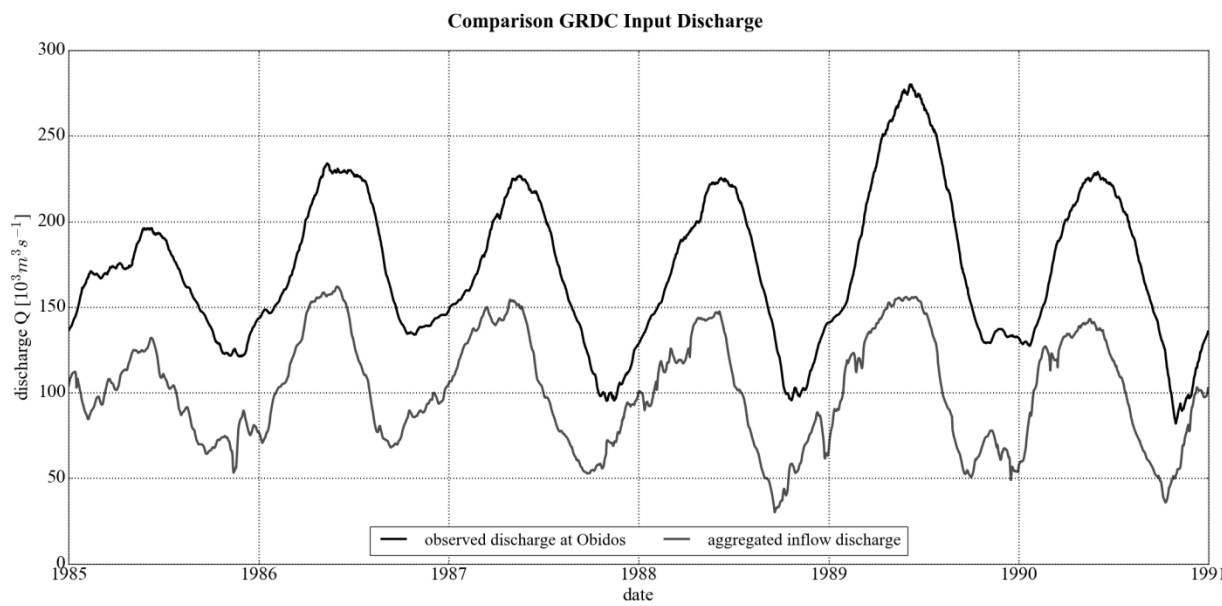

**Figure 6: Comparison of input discharge aggregated over all GRDC stations upstream of Óbidos and the observed discharge**

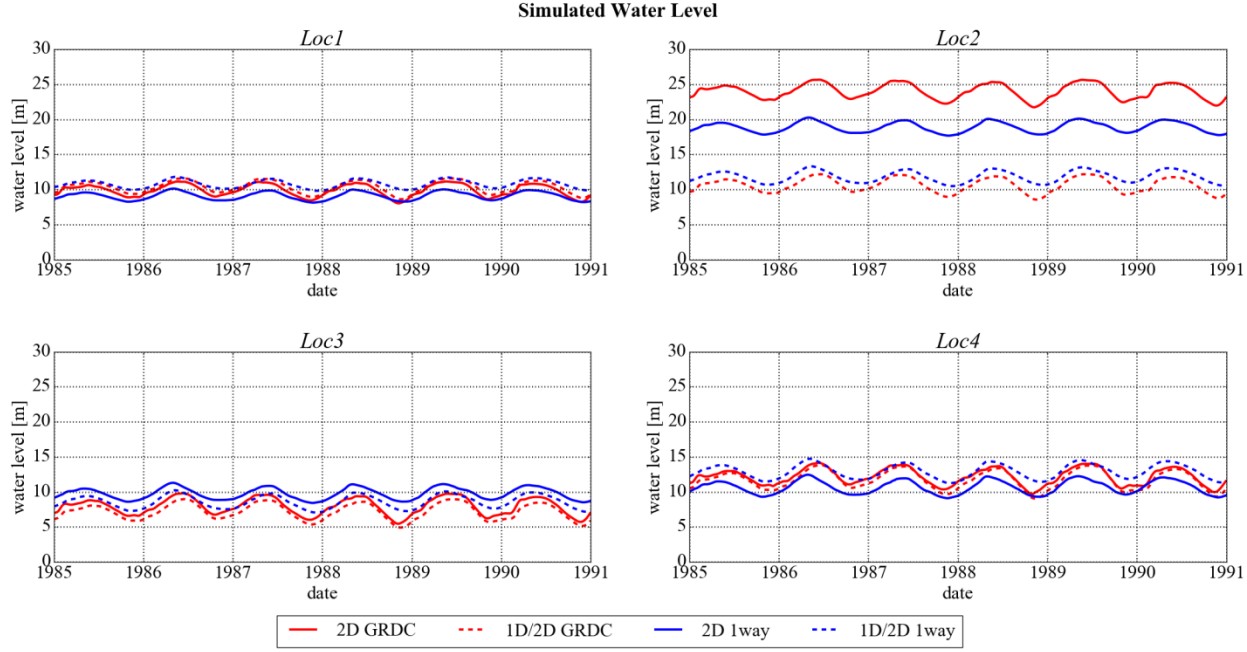

**Figure 7: Plot of simulated water levels at four different observation locations throughout the study domain**

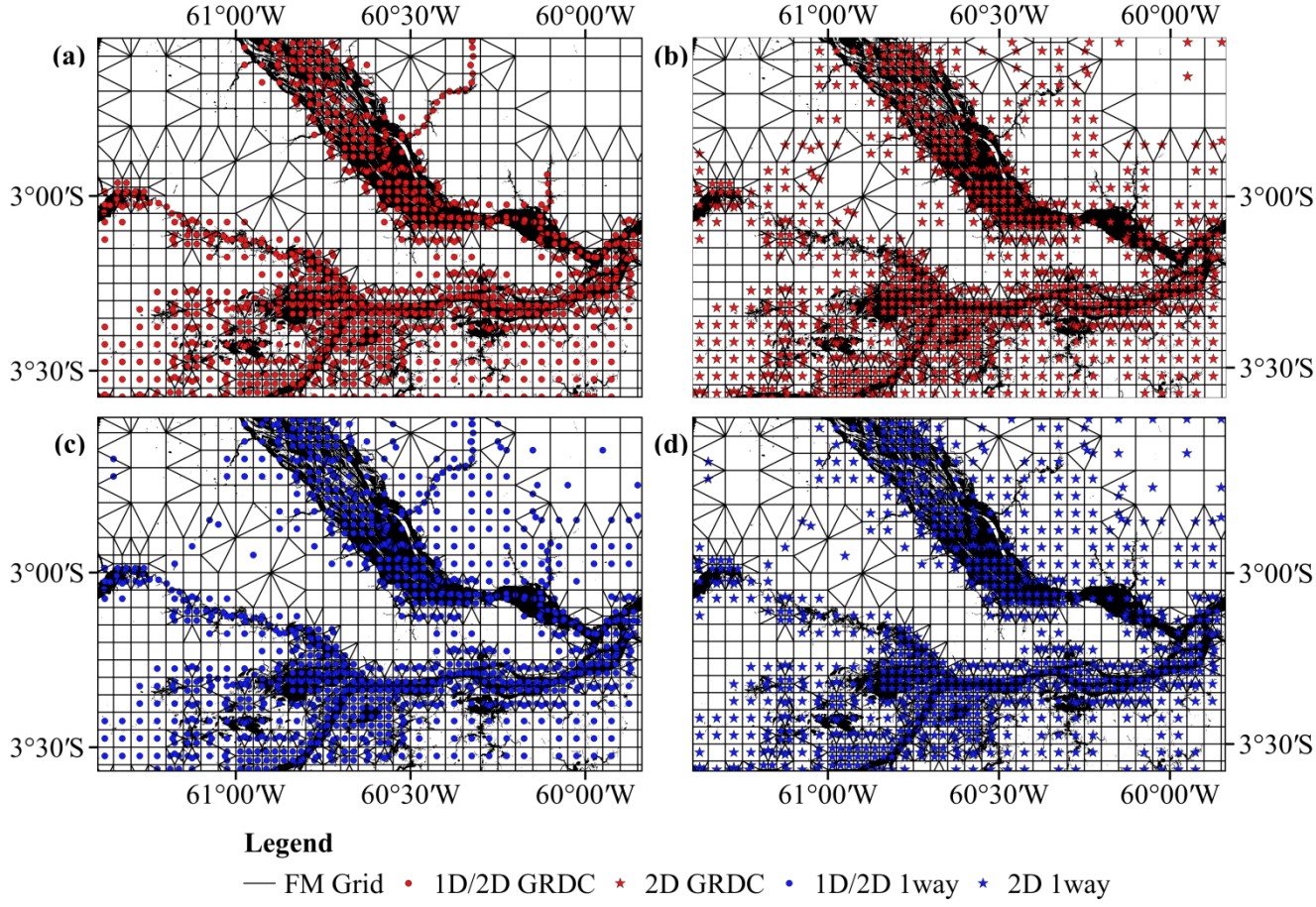

**Figure 8: Plot of simulated inundation extent per model set-up compared to observed water body extent as observed by LandSat imagery on July 1ˢᵗ 1989; the validation is performed for (a) the 1D/2D GRDC run, (b) the 2D GRDC run, (c) the 1D/2D coupled run, and (d) the 2D coupled run**

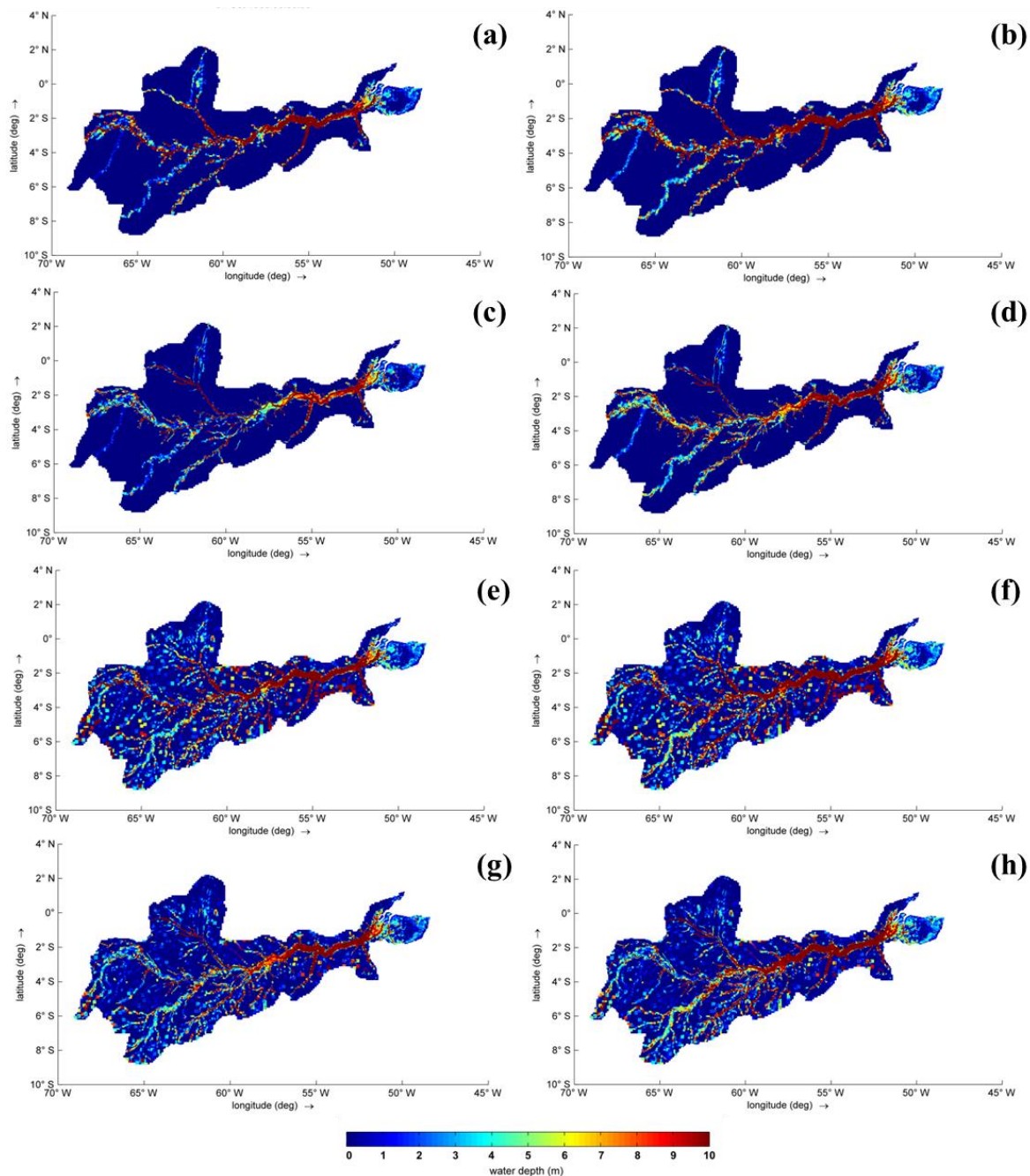

**Figure 9: Plots of simulated water depth for days with lowest (31 October 1990; left side) and highest discharge (08 June 1989; right side) at Óbidos; observed for 2D ((a) and (b)) and 1D/2D ((c) and (d)) with GRDC forcing; for 2D ((e) and (f)) and 1D2D ((g) and (h)) for 1way-coupled runs**

|  | DynRout | 2D GRDC | 1D/2D GRDC | 2D 1way | 1D/2D 1way |
|---|---|---|---|---|---|
| $r^2$ | 0.49 | 0.92 | 0.85 | 0.77 | 0.83 |
| RMSE | 34100 | 16229 | 18735 | 21451 | 19548 |
| KGE | 0.64 | 0.80 | 0.86 | 0.84 | 0.79 |

Table 1: Performance of model runs in SOFs for both actual and scaled model input

| Type | GRDC Station Name | GRDC Station Number | GRDC Catchment Area [km$^2$] |
|---|---|---|---|
| i | Caracarai | 3618500 | 124,980 |
| i | Uaracu | 3618950 | 40,506 |
| i | Acanaui | 3621200 | 242,259 |
| i | Sao Paulo de Olivenca | 3623100 | 990,781 |
| i | Gaviao | 3624120 | 162,000 |
| i | Aruma-Jusante | 3625310 | 359,853 |
| i | Porto Velho | 3627040 | 954,285 |
| i | Jiparana (Rondonia) | 3627408 | 32,606 |
| i | Estirao Da Angelica | 3628500 | 26,040 |
| Σ (catchment area input stations) | | | 2,933,310 |
| o | Óbidos | 3629000 | 4,640,300 |
| proportional representation of catchment area | | | 63% |

Table 2: List with catchment area per GRDC station located upstream of Óbidos (type 'i') compared to catchment area of observation station Óbidos (type 'o'); all data sourced from official GRDC website www. grdc.sr.unh.edu