# Peer review of "Assessing the impact of hydrodynamics on large-scale flood wave propagation – a case study for the Amazon Basin"

_Hydrology and Earth System Sciences, 2016_

## Referee Comment (RC1) · Anonymous Referee #1 · 22 Sep 2016

The paper investigates the benefits of coupling hydrological and hydrodynamic models to simulate large-scale flood propagation and inundation processes. The topic is very relevant and fits the journal's scope. Also, I found the paper well written and nice to read. Yet, I have two major concerns that I think should be addressed by the authors.

1) As models with more parameters obviously tend to fit data better, one of the traditional approaches to compare different model structures is to calibrate the alternative model configurations by using a set of data and then validate them against another (independent) set of data. It was not entirely clear to me if this was done here and how exactly model parameterisation was implemented. I think this is a crucial aspect because literature in both hydrological and hydraulic modelling has shown that the im-

pact of model parameters is often as significant as the impact of using specific model structures.

2) The focus of this case study is on flood inundation in a part of the Amazon basin, which is also the core of this modelling effort. Yet, numerical results are not compared in terms of flood extent. Isn't there any satellite data to get flood extent information? If not, is this the right case study to test this new methodology?

In conclusion, this is a very nice paper tackling a useful and timely issue, but it should: i) clarify the parameterization strategy and its impact on the conclusions as well as ii) justify the choice of the test site and the absence of a model comparison in terms of flood extent.

---

## Author Comment (AC1) · 3 Oct 2016

We thank the anonymous reviewer for his/her evaluation of our manuscript and helpful comments.

With regard to comment #1, we fully agree with the reviewer that it is necessary to evaluate the sensitivity to model parameters and to calibrate and validate them whenever possible. In this case, however, we investigate the improvements that can be obtained by using different hydrodynamic model set-ups when forced with simulated hydrology, in this case from the large-scale hydrological model PCR-GLOBWB. Thus, we opt to tune the hydrological output in terms of volume by optimizing at the basin scale a selection of five parameters in PCR-GLOBWB as outlined in the manuscript, resulting in

a 38% lower RMSE and a KGE improved by 68%. We do acknowledge that this procedure may not become perfectly clear in the manuscript, and will improve this in the revised manuscript. A further reason why we are reluctant to calibrate the model is that eventually we wish to apply our approach on the global scale. Local, expressly basin-scale- calibration would be desirable with the objective to improve forecast skills but it will introduce inconsistencies among the global datasets used and jeopardize their validity for ungauged basins. Bearing in mind the above, we fully subscribe to comment #2 and recognize that validating the modeled flood extent is of major importance; it adds an additional check on the validity of the different hydrodynamic models and tests the effects of wetted perimeter and resistance more directly than by validating against the observed discharge hydrograph only; also, observations of flood extent, e.g., by remote sensing, provide a denser global coverage than discharge observations. Thus, it offers a way to check the validity of the hydrodynamic models over larger areas, also in data scarce regions. At the time of writing, though, we chose not to validate against flood extent for two reasons. First, the spatial resolution of the hydrodynamic model is around 2.5km for the finest cells which introduces a mismatch when validated against finer-resolution satellite imagery that is directly due to resolution. This obscures the actual error that should be attributed to model structure and parameterization. Second, the temporal coverage of remotely sensed flooding may have gaps that further complicate a comparison with the continuously simulated flood extent. This issue can partly be addressed by matching the maximum flood extent over chosen periods (e.g., weeks, months . . .) but this will emphasize the effect of the local topography and dilute the ability to assess the skill of the different hydrodynamic models. Therefore we decided initially to concentrate on validating the simulated discharge at Óbidos. In light of the reviewer's comment we will try and accommodate a validation against the flood extent from remotely sensed data bearing in mind the limitations that we list here directly above. Depending on the quality and meaningfulness of the outcome, we will add this to the revised manuscript. In case quality and meaningfulness are considered to be insufficient, we will add this relevant validation step in an already started follow-up case

study that employs the coupling procedure at a much finer spatial resolution (< 1km). In any case, we expect the validation against flood extent to provide relevant information on the skill of the hydrodynamic models to simulate flood extent over space and time which cannot be gathered from a validation against discharge alone.
* * *

---

## Referee Comment (RC2) · D. Yamazaki (Referee) · 25 Oct 2016

This manuscript describes the differences of simulated river hydrodynamics among hydrology model, hydrodynamic model, and coupled model. The work is interesting because comparison of river routine framework has not been widely studied yet. Though some modifications are needed, I think this manuscript has a potential to be published on HESS after minor revision.

<Specific Comments>

P3.L32: 2D models experience problems in case the actual river width is smaller than the grid size and also in case there are multiple rivers within one cell, although it is

possible to partly overcome that by applying sub-gridding routines (Neal et al., 2012; Yamazaki et al., 2011).

> CaMa-Flood (Yamazaki et al., 2011) is a 1D global river model, so this description is not accurate.

> Please also note that MGB-IPH (Paiva et al., 2011) and CaMa-Flood (Yamazaki et al. 2011; 2013) are different from other 1D-hydrology and 2D-hydrodynamic models. They are 1D continental- or global-scale river models, but they utilize a shallow water equation as the governing flow equation. Other hydrology-type river models use kinematic-wave equations, and other 2D hydrodynamic models cannot easily be applied to continental scales. It is better to provide a careful review on these models.

P7.L6: River depth d [m] was subsequently estimated from river width w [m] by means of the following equations from Paiva et al.

> Is the river width calculated from drainage area? Or is it given by GWD-LR?

> The hydro-geometry equations (eq.1 and 2) are suitable for describing the general increase of the channel depth and width from upstream to downstream. However, if the width is given from observation (i.e. from GWD-LR), the equation (3) cannot be used to account the local variation of channel depth. In general, given that the discharge is same, channel is deeper when the width is smaller (and vice versa).

P8.L22: the finest spatial resolution (2.5 km × 2.5 km) for areas with lowest HAND values

> How the elevation of each 2D mesh is defined? Is it given as the average of 3sec pixels within the mesh? Or minimum elevation within the mesh? Please describe the detail because this could largely change the hydrodynamic simulations.

P8.L8: For the present study, river depth d was computed as a function of upstream area Ad

> Is this assumption consistent to the 1D/2D model? Given that the hydrodynamic simulation is very sensitive to channel bathymetry, we cannot rigorously compare the difference between the 2D and 1D/2D models if bathymetry is not consistent.

P9.L13: a delta volume was computed based on daily river discharge, surface runoff, and water layer volumes

> Please provide more detailed information on how to couple the hydrology and hydro-dynamic models. I guess, river discharge is used at the 2D-model's upstream boundaries, while surface runoff is used within the 2D-model's domain. But I'm not sure how the water layer volume is used.

P10.L3: Forcing the model with discharge observed at GRDC-stations, we found that the aggregated input discharge as obtained from upstream GRDC-station observations accounted for only 59% of the discharge generated in the basin as observed at Óbidos (Figure 5).

> Please describe the locations of GRDC stations used as upstream boundary input. Please also calculate how much percentage of the basin area is covered by the GRDC gauges. Without the above information, readers cannot understand the ∼30% under-estimation is reasonable or not.

P10.L13: the simulated discharge is consistently higher than of both the purely hydrology- and purely hydrodynamic-based models

> This is quite unusual, and I guess there is a possibility of a bug in the codes. The river routing scheme can alter the timing of hydrodynamics, but it does not change the total amount of flowing water (i.e. water mass is conserved).

> Therefore, one potential source is a loss of surface water (evaporation or infiltration). Please calculate the amount of surface water loss in PCR-GLOBWB, and confirm that the loss can explain the difference between the Hydro-only simulation and the coupled simulation.

> If the loss cannot explain the discrepancy, then please check the river network structure of each modelling framework. Especially in a coarse-resolution river network such as at 0.5 degree, the merging location of the mainstem and branches could be unrealistic.

> The error in mass balance calculation is critical, so that the cause of discrepancy should be examined more carefully.

P10.L34: A closer inspection of model results, nonetheless, reveals that the rate of increase as well as decrease of the rising and falling limb, respectively, is higher compared to the purely hydrology-based run.

> The rate of increase/decrease strongly depends on channel bathymetry. If the channel is deeper, the discharge increase faster, and vice versa. Therefore, the noted difference cannot be simply related to the way of coupling models.

P12.L18: We also found that GRDC-forced runs show stronger attenuation and lagged peak discharge due to the longer average travel time required to propagate from the boundaries through the model domain.

> Whether travel time becomes longer or shorter depends on the location of the input GRDC gauges. If the travel time could be longer if the missing input from neglected branches are located in downstream, but the travel time could be longer if neglected branches are in upstream.

Figure 7:

> Water depth is highly affected by local channel bathymetry. I think it is also better to compare the water surface elevation (above sea level), because water surface elevation is determined by larger-scale hydrodynamics.

---

## Author Comment (AC2) · 3 Nov 2016

We thank Dr. Dai Yamazaki for his detailed evaluation of our manuscript and helpful comments, which we will address stepwise here below.

With regard to your first comment on page3/line32 in the manuscript, we refer to CaMa-Flood's ability to determine water level, water storage, and flooded area on basis of sub-grid topography. We acknowledge that this is not precisely equivalent to a sub-grid channel representation as discussed in the manuscript, and thank Dr. Yamazaki for noticing these relevant differences. We will update this in the revised manuscript.

Answering your comment on page7/line6, we thank you for raising some relevant as-

pects regarding the computation of the river bathymetry. Your latter assumption holds, as we computed river depth based on river width taken from GWD-LR and estimated the river depth with Equation 3. This relationship is therefore independent of upstream area. We preferred to use the width from GWD-LR as it better reflects the local conditions than the generic relationship of Paiva et al. (2011) in which only upstream area is the predicting variable, even though the latter was derived for the Amazon. We believe that estimating river depth from river width with Equation 3 holds as the formulas – although empirical in nature – are internally consistent, and moreover derived from data collected at 341 cross-sections (see Paiva et al. (2011), therefore already gaining more general validity. Also, bathymetric information is fed into the model as averages for segments between profile locations, and thus local variability in width and depth is neglected, and the influence of local river restrictions or widenings diminished. Similar to our approach, Neal et al. (2012) estimated river depth from observed width by smoothing river widths and thereafter re-shuffling the hydro-geometric equations of Leopold and Maddock (1953). These approaches have the advantages that they remove the need to include an estimate of bankful discharge or upstream area a priori, and that an educated estimate of river bathymetry can be obtained where no information is available. In a nutshell, we agree with your valid comment that basing the relationship directly on observed width does not ensure downstream continuity as merely using the discharge or the upstream area would. However, we believe that the advantage of using observed widths in combination with estimated depths outweigh this theoretical limitation due to the arguments listed above.

Regarding your comment on page8/line 22, the elevation of each flexible mesh cell is computed as the normal unweighted average of the elevations assigned to each cell vertex. The cell vertices have their elevation assigned by employing a sample-file in ascii-format containing gridded elevation data. We will have improved the description of the method and added the relevant details in the revised manuscript.

In answer to your comment on page8/line8, we indeed used the same equations and

assumptions for defining bathymetry in the 1D/2D and 2D schematization, yet in a slightly different manner. While for 1D/2D we ignored upstream area and derived depth directly from river width, we lacked river width in 2D and therefore estimated river depth from upstream area. A systematic comparison between computed depths was not possible as the 1D-network does not necessarily coincide with the LDD use to determine upstream area. However, a manual inspection of points where this was the case showed that differences between obtained depth for 1D/2D and 2D are within of 10%. We therefore are confident that these deviations in methodology do not have a strong impact on model behaviour at the large-scale which is underpinned by experiences made in other case studies. We nevertheless consent that these inequalities, altogether with the still present differences between the set-ups, need to be pronounced stronger in the final manuscript to avoid any lack of clarity.

With regard to your comment on page9/line 13, we understand that the coupling-algorithm is not perfectly clear in its explanation, and thus needs some details added. Indeed, your assumption is correct. With the discharge volumes being the result of PCR-GLOBWB routing upstream of the area where coupling is active, and hence being transferred to the hydrodynamic model only at the boundary cells, surface runoff is used as input to the hydrodynamic model over the coupled domain instead of being fed into the PCR-GLOBWB routing scheme. The "TopWaterLayer" is an additional layer that can be populated with inundation depth information from the hydrodynamic model. Therefore, this layer will only be influential once two-way coupling is performed, and due to that we will remove the layer from both flowchart and text to highlight the other, more important data and processes.

Regarding your comment on page10/line3, we will enhance the specific section to improve the reader's understanding. We will add the GRDC-stations in the plot with river depth observation stations and provide a table showing the specific properties of each station. With this additional information, we are confident that the under-representation of discharge by GRDC-stations will become better understandable.

With respect to your comment on page10/line13, we express our agreement with your statement that different routing schemes may result in different timing of flood wave occurrence as well as possible attenuation effects. Over longer simulations as ours, the water balance should be closed, and the water volumes added to the model should equal the outflow volumes. Since a closed water balance is pivotal for meaningful hydrologic and hydrodynamic computations, we re-assure you that we double-checked it. Our test runs showed that, first, the hydrodynamic model is mass conservative, that is no water is lost or added, which is, given that the model is designed for commercial applications too, a prerequisite. And second, we assessed whether the input volumes (river discharge, surface runoff, and topWaterLayer) from PCR-GLOBWB which are added to Delft3D FM as delta volumes equal the volumes as computed by PCR-GLOBWB without coupling. The answer is yes, they are equal. As a result, we conclude that the amounts of water per model run are correct, but found other causes for differences in observed discharge. Reading the manuscript again, we find that these reasons may not have been explained clearly enough. It can indeed be the case that a model run shows higher or lower discharge values than others, but not because the water balances are incorrect. The reason is that we compare different model set-ups (hydrology-only, hydrodynamics-only in 1D/2D and 2D, coupled hydrology-hydrodynamics in 1D/2D and 2D) which can different rather remarkably between each other. While the discharge timing and shape are governed by the routing module applied, the total volumes are controlled by the hydrologic processes represented. The fact that discharge in coupled runs is consistently higher than other runs may be because of (a) the hydrodynamic-only runs are forced with data from different origin (GRDC), and (b) the hydrology-only runs are forced with data from same origin (PCR-GLOBWB), but account for groundwater infiltration and evaporation processes on all water bodies in the model domain, which lowers the overall simulated discharge. In the final manuscript, we will take your valuable comment into consideration, and add information where necessary to ensure the different discharge results are not taken as result of water balance errors.

To react to your comment on page10/line34, we are thankful that you pointed towards this indistinctness. It is entirely true that not the way of coupling, but mainly the river bathymetry is decisive when it comes to simulating the timing as well dynamics of flood waves. We do not aim to attribute the differences in discharge dynamics to the fact that we coupled two models, but to the choice of model used to simulate discharge and inundations (hydrology-only, hydrodynamics-only in 1D/2D and 2D, coupled hydrology-hydrodynamics in 1D/2D and 2D). It is inherent to the various model set-ups that some properties such as for instance bathymetry differ between them. For the revised manuscript, we will enhance the text accordingly.

Commenting on your remarks to page 12/line18, we are entirely in line with you and agree that the timing of discharge depends on the distance between input and observation station. Our way to account for 'missing', that is underrepresented, discharge volumes contains a uniform scaling factor which is applied to each input location, hence not altering the relative contributions of each inflow. Thus, we think our conclusion that using input at a model domain's edge leads to peak attenuation and lag still holds true.

With respect to your comment on Figure 7, we concur with your remark. For the present manuscript, we decided to plot water depth to indeed highlight the impact of local bathymetry and therefore the impact of general model set-up differences between 2D and 1D/2D. In order to keep this argument, but also to display the important larger-scale hydrodynamics, we will extent the manuscript with a plot of water elevation above sea level and the related discussion in the text.
* * *

---

## Author Response (AR1)

*We thank referee #1 for the kind words on our manuscript and the points brought forward as they resulted in an improvement of the submitted manuscript. We have, in particular, added a validation of simulated inundation which we think highly improves the quality of the study. Below, we repeat the reviewer's comments, and provide our response in italics.*

As models with more parameters obviously tend to fit data better, one of the traditional approaches to compare different model structures is to calibrate the alternative model configurations by using a set of data and then validate them against another (independent) set of data. It was not entirely clear to me if this was done here and how exactly model parameterisation was implemented. I think this is a crucial aspect because literature in both hydrological and hydraulic

10 modelling has shown that the impact of model parameters is often as significant as the impact of using specific model structures.

*With regard to comment #1, we fully agree with the reviewer that it is necessary to evaluate the sensitivity to model parameters and to calibrate and validate them whenever possible. In the present study, however, we aim to investigate the improvements that can be obtained by using different hydrodynamic model set-ups when forced with simulated hydrology, in*

15 *this case from the large-scale hydrological model PCR-GLOBWB, compared to using observed boundary discharge.*

*To allow better comparability with observed discharge values as well as simulated discharge values from runs with upstream boundary forcing, we tuned the hydrological output in terms of volume by optimizing at the basin scale a selection of five parameters in PCR-GLOBWB. Such tuning is required as it results in a 38% lower RMSE and a KGE improved by 68% of simulated PCR-GLOBWB discharge. We do acknowledge that the motivation and steps taken may not become*

20 *perfectly clear in the manuscript, and have elaborated on the reasoning as well process description in chapter 2.1 to provide clearer understanding to the reader.*

*We desisted from a more elaborated model parameter calibration as we eventually wish to apply our approach on the global scale. Local, expressly basin-scale- calibration would be desirable with the objective to improve forecast skills but it will introduce inconsistencies among the global datasets used, and jeopardize their validity for ungauged basins. We pointed*

25 *towards these aspects and explained our decisions more clearly by adding remarks where necessary, that is sections 2.1 and 2.2.*

The focus of this case study is on flood inundation in a part of the Amazon basin, which is also the core of this modelling effort. Yet, numerical results are not compared in terms of flood extent. Isn't there any satellite data to get flood extent

30 information? If not, is this the right case study to test this new methodology?

*Given the importance of validation of inundation extent as also rightly stressed by the reviewer, we are very happy to have added a first-order and qualitative validation which we think contributes strongly to the overall quality and soundness of the study. To do so, we employed LandSat imagery taken on July 1st 1989 in the area of Manaus between around 60°00' W and 61° 30' W and 2°30' S and 3°30' S and validated all results obtained with Delft3D Flexible Mesh, hence not the result*

*obtained with PCR-GLOBWB-DynRout due to its 30 arc minutes grid which makes it unfit for this purpose. All meshes containing a water depth value have been defined as flooded, and subsequently overlaid with the LandSat imagery to allow for a first-order assessment of inundation extent. The results of this assessment showed generally good agreement, with less accurate results for coupled runs in floodplain areas, and improved performance for all runs employing 1D channels compared to 2D only. We added these findings in the results section as well as allocated a part of the discussion to the validated inundation extent. For visualization purposes, we added a plot of observed and simulated extent as well (Figure 8 in revised manuscript).*

*Despite this showing the skill and potential of the presented model set-up, we are going to repeat this validation step in an already started follow-up case study that employs the coupling procedure at a much finer spatial resolution (< 1km).*

*With the additions made to the manuscript based on the valuable reviewer's remarks, we are convinced to have responsibly addressed all uncertainties, ambiguities, and shortcomings of the initially submitted version.*

*We thank referee #2, Dr. Dai Yamazaki, for the kind words on our manuscript and the points brought forward as they resulted in an improvement of the submitted manuscript. Below, we repeat the reviewer's comments, and provide our response in italics.*

P3.L32: 2D models experience problems in case the actual river width is smaller than the grid size and also in case there are multiple rivers within one cell, although it is possible to partly overcome that by applying sub-gridding routines (Neal et al., 2012; Yamazaki et al., 2011).

> CaMa-Flood (Yamazaki et al., 2011) is a 1D global river model, so this description is not accurate.

10 > Please also note that MGB-IPH (Paiva et al., 2011) and CaMa-Flood (Yamazaki et al. 2011; 2013) are different from other 1D-hydrology and 2D-hydrodynamic models. They are 1D continental- or global-scale river models, but they utilize a shallow water equation as the governing flow equation. Other hydrology-type river models use kinematic-wave equations, and other 2D hydrodynamic models cannot easily be applied to continental scales. It is better to provide a careful review on these models.

15 *Due to the very helpful remark made, we have rectified this by removing the citation from the list. Regarding the review of the discussion on global or continental models, we have desisted from adding additional text to keep the balance of the manuscript and avoid prolonging the introduction, despite recognizing the added scientific value of both models in the field of hydraulic modelling.*

20 P7.L6: River depth d [m] was subsequently estimated from river width w [m] by means of the following equations from Paiva et al.

> Is the river width calculated from drainage area? Or is it given by GWD-LR?

> The hydro-geometry equations (eq.1 and 2) are suitable for describing the general increase of the channel depth and width from upstream to downstream. However, if the width is given from observation (i.e. from GWD-LR), the equation (3) cannot

25 be used to account the local variation of channel depth. In general, given that the discharge is same, channel is deeper when the width is smaller (and vice versa).

*We are thankful for raising these concerns, but, although understanding the concerns raised, we preferred to use the width from GWD-LR for input to Eq. 3 as it better reflects the local conditions than the generic relationship of Paiva et al. (2011) in which only upstream area is the predicting variable. We believe that our approach holds as the formulas – although*

30 *empirical in nature – are internally consistent, and moreover generically enough as they are derived from 341 cross-sections (see Paiva et al. (2011). In addition, bathymetric information is fed into the model as averages for segments between profile locations, and thus local variability in width and depth is neglected. The fact that other studies, for instance Neal et al. (2012), employed comparable approaches gives our approach another layer of validity.*

*Based on these arguments, we believe that the advantage of using observed widths in combination with estimated depths is sufficiently underpinned. Hence, we did not change our methodology, but added extra text in section 2.3 merely dedicated to the limitations and concerns rightfully raised by the reviewer regarding the approach chosen for improved clarity.*

P8.L22: the finest spatial resolution (2.5 km × 2.5 km) for areas with lowest HAND values

> How the elevation of each 2D mesh is defined? Is it given as the average of 3sec pixels within the mesh? Or minimum elevation within the mesh? Please describe the detail because this could largely change the hydrodynamic simulations.

*We have added a short description to section 2.4 to make the internal computations better understandable.*

P8.L8: For the present study, river depth d was computed as a function of upstream area Ad

> Is this assumption consistent to the 1D/2D model? Given that the hydrodynamic simulation is very sensitive to channel bathymetry, we cannot rigorously compare the difference between the 2D and 1D/2D models if bathymetry is not consistent.

*Thanks to this highly constructive comment, we assessed differences in computed river depth. Albeit the differences inherent in schematization of 1D/2D and 2D set-ups, we could detect only minor differences in computed river depth. In light of these minor differences, we deem the differences in schematization acceptable as we think that the approach chosen is the best approach possible, also given the data and data types to be used. Since the limitations of the approach are noteworthy, as also indicated by the reviewer's comment, we added extra text in section 2.4 to better explain the choices made, and to stronger pronounce the differences and limitations, promoting clarity of steps taken.*

P9.L13: a delta volume was computed based on daily river discharge, surface runoff, and water layer volumes

> Please provide more detailed information on how to couple the hydrology and hydrodynamic models. I guess, river discharge is used at the 2D-model's upstream boundaries, while surface runoff is used within the 2D-model's domain. But I'm not sure how the water layer volume is used.

*As it is rightly stated by the reviewer, the role of the "topWaterLayer" is rather dubious in the manuscript. For the plot (Figure 5 in revised manuscript) and section 2.5, we have removed the "water layer" as the volumes obtained from this variable are zero in one-directional coupling and only become active once two-directional coupling as activated which is not the case in this study but already implemented for future applications.*

P10.L3: Forcing the model with discharge observed at GRDC-stations, we found that the aggregated input discharge as obtained from upstream GRDC-station observations accounted for only 59% of the discharge generated in the basin as observed at Óbidos (Figure 5).

> Please describe the locations of GRDC stations used as upstream boundary input. Please also calculate how much percentage of the basin area is covered by the GRDC gauges. Without the above information, readers cannot understand the ~30% underestimation is reasonable or not.

*To make the underrepresentation of discharge at Obidos better tangible, we calculated the upstream area represented by each GRDC-station upstream of Obidos and compare the summed upstream area to the area represented by Obidos itself. We found that the summed upstream area can only account for 62% of area represented by Obidos itself which is very much in line with the 59% of represented discharge. Thus, this finding can be used to explain the lack of inflow discharge volumes*

5 *by GRDC-stations upstream of Obidos compared to observed discharge at Obidos itself. To make this better understandable for the reader, we added the locations of all GRDC-stations as well as the area of the entire Amazon Basin to Figure 3 in the revised manuscript. Besides, we added a table to show the actual numbers and computations made to derive the upstream areas, along with the GRDC-station identifiers for traceability. In section 3.1, we also added information and links to the figure and table.*

P10.L13: the simulated discharge is consistently higher than of both the purely hydrology- and purely hydrodynamic-based models

> This is quite unusual, and I guess there is a possibility of a bug in the codes. The river routing scheme can alter the timing of hydrodynamics, but it does not change the total amount of flowing water (i.e. water mass is conserved).

15 > Therefore, one potential source is a loss of surface water (evaporation or infiltration). Please calculate the amount of surface water loss in PCR-GLOBWB, and confirm that the loss can explain the difference between the Hydro-only simulation and the coupled simulation.

> If the loss cannot explain the discrepancy, then please check the river network structure of each modelling framework. Especially in a coarse-resolution river network such as at 0.5 degree, the merging location of the mainstem and branches

20 could be unrealistic.

> The error in mass balance calculation is critical, so that the cause of discrepancy should be examined more carefully.

*We very much acknowledge the importance of the topic raised by the reviewer, and therefore have double-checked the water balance again, and again could not detect any bugs in the coupling code. We are hence sure that there are no water balance errors in our model results. The discrepancies between simulated discharge of each set-up is therefore entirely due*

25 *differences in processes modelled (coupled runs don't account for evaporation and groundwater infiltration in contrast to hydrology-only run), scale (hydrology-only in contrast to coupled runs), and water volumes used for model forcing (GRDC-runs in contrast to coupled run). As we had already identified these aspects as causes in the previous manuscript, we elaborated on the text in section 3.1 to better clarify the differences in schematizations and processes implemented, as well as their implications for simulated discharge.*

P10.L34: A closer inspection of model results, nonetheless, reveals that the rate of increase as well as decrease of the rising and falling limb, respectively, is higher compared to the purely hydrology-based run.

> The rate of increase/decrease strongly depends on channel bathymetry. If the channel is deeper, the discharge increase faster, and vice versa. Therefore, the noted difference cannot be simply related to the way of coupling models.

*As consequence of the comment made, we re-assessed model results, and came to the conclusion that, after closer examination, the finding made no longer holds as we found that the rate of increase and decrease do not remarkably differ between model set-up. As a result, we removed the text passage from the manuscript and have re-written section 3.1 accordingly.*

P12.L18: We also found that GRDC-forced runs show stronger attenuation and lagged peak discharge due to the longer average travel time required to propagate from the boundaries through the model domain.

> Whether travel time becomes longer or shorter depends on the location of the input GRDC gauges. If the travel time could be longer if the missing input from neglected branches are located in downstream, but the travel time could be longer if

10 neglected branches are in upstream.

*While agreeing with your remark, our approach of applying a uniform scaling factor does not impose any conflicts with locations of branch locations, and therefore we did not change the manuscript.*

Figure 7:

15 > Water depth is highly affected by local channel bathymetry. I think it is also better to compare the water surface elevation (above sea level), because water surface elevation is determined by larger-scale hydrodynamics.

*As we do agree with your remark, we decided to replace water depth with water level to better show large-scale hydrodynamics. Therefore, we have changed the figure accordingly, and also re-written the related section 3.2*

*With the additions made to the manuscript based on the valuable and critical reviewer's remarks, we are convinced to have responsibly addressed all uncertainties, ambiguities, and shortcomings of the initially submitted version.*

**_Main Changes to Manuscript_**

For better overview in the marked-up manuscript, all changes made in relation to the comments of the reviewers are additionally highlighted yellow.

Comments Reviewer #1

1. Added extra text to clarify the model parameterizations of PCR-GLOBWB and Delft3D Flexible Mesh better and provided reasoning for not performing any calibrations of model parameters (sections 2.1 and 2.2)
2. Performed validation of flood extent by comparing results to remotely sensed imagery. Added paragraph accordingly (section 3.2) and plot of simulated versus observed water extent.

Comments Reviewer # 2

1. Page3/Line32: Adapted citations accordingly to fit with comment
2. Page7/Line6: Elaborated and briefly discussed our reasoning why using hydro-geomorphic relations is valid for large-scale applications and in light of the data and models we use (section 2.3)
3. Page8/Line22: Added text on how elevation values are assigned to flexible mesh of Delft3D FM (section 2.4)
4. Page8/Line8: Elaborated on the choices made to use the approach under discussion (section 2.4)
5. Page9/Line13: Removed all links to "topWaterLayer" and updated text as well as plot accordingly (section 2.5); also elaborated more on coupling procedure in general
6. Page10/Line3: added locations of inflow stations to Figure 1, and added table with upstream areas per location (Table 1); adapted text accordingly to better show cause for underrepresentation of discharge (section 3.1)
7. Page10/Line13: Re-written parts of section to better point out reason for differences in simulated discharge (section 3.1)
8. Page10/Line34: Removed passage as not agreeing with it anymore after closer examination of discharge results; rewritten text accordingly (section 3.1)
9. Page12/Line18: No changes made to manuscript
10. Figure7: Assessed water levels instead of water depth; adapted results and conclusion as well as plot accordingly (section 3.2)

[revised manuscript text omitted]